# A Simple and Direct Assay for Monitoring Fatty Acid Synthase Activity and Product-Specificity by High-Resolution Mass Spectrometry

**DOI:** 10.3390/biom10010118

**Published:** 2020-01-10

**Authors:** Magdalena Topolska, Fernando Martínez-Montañés, Christer S. Ejsing

**Affiliations:** 1Department of Biochemistry and Molecular Biology, Villum Center for Bioanalytical Sciences, University of Southern Denmark, 5000 Odense, Denmark; magdalenat@bmb.sdu.dk (M.T.); fernandomm@bmb.sdu.dk (F.M.-M.); 2Cell Biology and Biophysics Unit, European Molecular Biology Laboratory, 69117 Heidelberg, Germany

**Keywords:** fatty acid synthase, enzyme activity, high-resolution shotgun lipidomics, Orbitrap mass spectrometry

## Abstract

*De novo* fatty acid synthesis is a pivotal enzymatic process in all eukaryotic organisms. It is involved in the conversion of glucose and other nutrients to fatty acyl (FA) chains, that cells use as building blocks for membranes, energy storage, and signaling molecules. Central to this multistep enzymatic process is the cytosolic type I fatty acid synthase complex (FASN) which in mammals produces, according to biochemical textbooks, primarily non-esterified palmitic acid (NEFA 16:0). The activity of FASN is commonly measured using a spectrophotometry-based assay that monitors the consumption of the reactant NADPH. This assay is indirect, can be biased by interfering processes that use NADPH, and cannot report the NEFA chain-length produced by FASN. To circumvent these analytical caveats, we developed a simple mass spectrometry-based assay that affords monitoring of FASN activity and its product-specificity. In this assay (i) purified FASN is incubated with ^13^C-labeled malonyl-CoA, acetyl-CoA, and NADPH, (ii) at defined time points the reaction mixture is spiked with an internal NEFA standard and extracted, and (iii) the extract is analyzed directly, without vacuum evaporation and chemical derivatization, by direct-infusion high-resolution mass spectrometry in negative ion mode. This assay supports essentially noise-free detection and absolute quantification of *de novo* synthetized ^13^C-labled NEFAs. We demonstrate the efficacy of our assay by determining the specific activity of purified cow FASN and show that in addition to the canonical NEFA 16:0 this enzyme also produces NEFA 12:0, 14:0, 18:0, and 20:0. We note that our assay is generic and can be carried out using commonly available high-resolution mass spectrometers with a resolving power as low as 95,000. We deem that our simple assay could be used as high-throughput screening technology for developing potent FASN inhibitors and for enzyme engineering aimed at modulating the activity and the product-landscape of fatty acid synthases.

## 1. Introduction

In eukaryotic cells, fatty acid biosynthesis is carried out via a conserved polymerization reaction where the canonical substrates acetyl-CoA, malonyl-CoA, and NADPH are used to produce fatty acyls (FAs) of varying chain lengths [1,2]. These FA chains are used as building blocks for making biological membranes, for energy storage, and for making signaling molecules [3]. Dysfunctional regulation of fatty acid biosynthesis is implicated in numerous diseases, including cancer [4,5,6], type-2-diabetes [7,8], and cardiovascular diseases [9,10,11]. Due to this, research has for the last century been focused on understanding the molecular underpinnings of fatty acid biosynthesis and combating its detrimental effects when dysregulated. More recently, and important in the wake of the world’s climate crisis, biotechnological research is nowadays also focused on repurposing and optimizing *de novo* fatty acid synthesis for large-scale production of sustainable biofuels [12,13,14].

*De novo* fatty acid biosynthesis is catalyzed by fatty acid synthases (FASs). The animal and fungal FASs are among the best understood and belong to the so-called type I FAS system. These enzyme systems are large macromolecular assemblies with catalytic domains are distributed across one or two polypeptide chains [3]. The mammalian FAS (FASN) consists of two 270 kDa α-polypeptide chains forming an α_2_-homodimer complex [1,15]. In the fungal FAS, the catalytic domains are distributed across two polypeptides, integrated into a 2.6 MDa α_6_β_6_-dodecamer complex [15,16,17]. In the initial step of *de novo* fatty acid biosynthesis (Appendix A), the acetyl-moiety from acetyl-CoA is transferred to a pantetheine residue in the mobile acyl-carrier protein (ACP) domain, a transfer that is catalyzed by the malonyl-acetyl-transferase (MAT) domain. Next, the acetyl-moiety is transferred to the ketoacyl synthase (KS) domain, which makes the ACP domain able to accept the malonyl-moiety from malonyl-CoA, a process that is also catalyzed by the MAT domain. Subsequently, the acetyl- and malonyl-moieties are condensed by the KS to produce a ketobutyryl-group, which is attached to the mobile ACP domain. Next, this intermediate is moved to the ketoacyl reductase [18] domain, where it becomes reduced using NADPH; then to the dehydratase (DH) domain, where it becomes dehydrated; and subsequently to the enoyl reductase (ER), where it becomes reduced using NADPH to yield a butyryl-group (FA 4:0). This butyryl-group is transferred back to the KS domain such that the ACP domain can accept another malonyl-moiety and the reaction cycle can be repeated several times whereby two carbon units are progressively added to the growing FA chain. In the mammalian FASN complex, this polymerization reaction is most frequently terminated when the FA length reaches 16 carbon atoms and is released from the enzyme as a non-esterified fatty acid (NEFA) by the thioesterase (TE) domain.

Assays for monitoring FAS activity can be divided into three main types: (i) the classic and user-friendly spectrophotometry-based assay that monitors NADPH consumption at 340 nm [19]; (ii) radioactivity-based assays that track the incorporation of ^14^C- or ^3^H-labeled acetyl-CoA or malonyl-CoA by liquid scintillation counting [20,21,22]; and (iii) mass spectrometry-based assays that monitors the incorporation of stable isotope labeled acetyl-CoA and/or malonyl-CoA [23,24,25]. The first two approaches are indirect assays as they fail to report the chain-length of *de novo* synthetized FA chains. Moreover, the spectrophotometry-based assay can be biased by interfering processes that use NADPH, and the radioactive assays are typically associated with excessive safety regulations that limits analytical throughput. In comparison, mass spectrometry-based assays afford direct and specific monitoring of *de novo* synthetized stable isotope-labeled FA analytes. To date, these assays all use gas chromatography-mass spectrometry (GC-MS), which require chemical derivatization of NEFA analytes, a relatively long analysis time (at least 15 min), and the use of multiple isotope-labeled FA standards and generation of multiple calibration curves for accurate quantification of *de novo* synthetized FA chains having different chain-length. Notably, we recently demonstrated that total FA analysis, which is traditionally done by GC-MS, can also be performed very easily and accurately by direct-infusion high-resolution MS analysis in 1 min [26]. Furthermore, we also showed that this approach can specifically monitor stable isotope-labeled palmitoyl chain (FA 16:0 (+^13^C_16_)) in plasma lipids from normoinsulinemic and hyperinsulinemic human subjects. This suggested to us that this simple MS-based approach could potentially also be used as specific readout for determining FASN activity.

Here, we describe a novel high-resolution MS-based assay for monitoring FASN activity in vitro. The method is based on incubating purified FASN with ^13^C_3_-malonyl-CoA, acetyl-CoA and NADPH, followed by direct quantitative monitoring of *de novo* synthesized ^13^C_3_-labeled products by direct-infusion (shotgun) high-resolution MS. We show that the approach supports specific detection and accurate absolute quantification of newly synthesized ^13^C-labeled NEFAs. Furthermore, we also show that cow FASN in addition to the canonical NEFA 16:0 also produces NEFA 12:0, 14:0, 18:0, and 20:0. We deem that the analytical hallmarks of our assay, together with the relatively simple sample preparation routine, the short analysis time, and the ease of automation, could make our assay useful for high-throughput screening technology designed for development of potent FASN inhibitors and for enzyme engineering aimed at modulating the activity and the product-landscape of FASs.

## 2. Materials and Methods

### 2.1. Chemicals and Standards

Methanol, *n*-hexane, 2-propanol, and water were purchased from Bisolve BV (Valkenswaard, Netherlands). Chloroform was from Rathburn Chemicals (Walkerburn, UK). Ammonium formate was purchased from Sigma-Aldrich (Buchs, Switzerland). The standard NEFA 16:0(+^2^H_4_) was purchased from Larodan Fine Chemicals (Solna, Sweden). The standard 19:0-CoA was purchased from Avanti Polar Lipids (Alabaster, AL, USA). All other chemicals, including ^13^C_3_-malonyl-CoA, acetyl-CoA, NADPH, cysteine, and methylamine, were purchased from Sigma-Aldrich (Merck, Darmstadt, Germany). All solvents and chemicals were HPLC grade.

### 2.2. FASN Activity Assay

Purified cow FASN was kindly provided by Nils J. Færgeman (University of Southern Denmark). FASN was purified from cow mammary gland by precipitation with ammonium sulfate, followed by purification using ion exchange chromatography and gel filtration chromatography [27]. Protein concentration was determined using the Pierce^TM^ BCA Protein Assay Kit (Pierce Biotechnology, Rockford, IL, USA). The FASN activity assay was carried in 2 mL vials at 37 °C in a total reaction volume of 200 µL, containing 100 mM potassium phosphate (pH 6.5), 2 mM EDTA, 300 µg/mL fatty acid free BSA, 10 mM cysteine, 200 µM NADPH, 16.5 µg purified FASN (stored in 100 mM potassium phosphate (pH 6.5)), 50 µM acetyl-CoA, and 80 µM malonyl-CoA. The cysteine solution was prepared prior to the assay. Samples were collected at indicated time points and the reactions were terminated by adding the reaction mixtures to vials containing solvents for lipid extraction (described in further detail below).

### 2.3. Sample Preparation

For monitoring ^13^C-labeled NEFAs (Figure 1) we initially made use of the sample preparation routine described by Gallego et al. [26]. In short, the reaction mixture (200 µL) was mixed with 100 µL 5 M HCl containing 150 pmol of the internal standard 19:0-CoA. This sample was mixed on a ThermoMixer (Eppendorf, Wesseling-Berzdorf, Germany) at 700 rpm for 2 h at 90 °C. After cooling, NEFAs were extracted by adding 700 µL of hexane and mixing at 1400 rpm for 10 min at 4 °C. The extraction with hexane was repeated, and the combined extract was vacuum evaporated. The final extract was dissolved in 150 µL chloroform/methanol (1:2, *v*/*v*) prior to mass analysis.

For optimization of the workflow (Figure 2), we tested four different sample preparation methods: (i) we used the above-described routine, but spiked samples with 215 pmol of the internal standard NEFA 16:0(+^2^H_4_), extracted samples with 700 µL of *n*-hexane, and dissolved the extract in 100 µL chloroform/methanol (1:2, *v*/*v*); (ii) we directly extracted the 200 µL reaction mixture with 700 µL of *n*-hexane containing 215 pmol of NEFA 16:0(+^2^H_4_) standard (no addition of 5 M HCl), and dissolved the final extract in 100 µL chloroform/methanol (1:2, *v*/*v*); (iii) we used a modified Bligh and Dyer [28] procedure where the 200 µL reaction mixture was spiked with 215 pmol NEFA 16:0(+^2^H_4_) standard, added 990 µL chloroform/methanol (2:1, *v*/*v*), and mixed the sample on a ThermoMixer at 1400 rpm for 1 h at 4 °C. The resulting extract was vacuum evaporated and dissolved in 100 µL chloroform/methanol (1:2, *v*/*v*); and (iv) we used the modified Bligh and Dyer procedure but omitted the vacuum evaporation step and instead analyzed the crude extract directly after a 4.5 min centrifugation at 13,000× *g* at 4 °C.

### 2.4. Mass Spectrometric Analysis

Extracts (11.7 µL) obtained using hexane-based extraction were loaded in a 96-well plate (Eppendorf, Hamburg, Germany) together with 15 µL 1.33 mM ammonium formate in 2-propanol. Extracts (15 µL) obtained using chloroform-based extraction were loaded in a 96-well plate (Eppendorf, Hamburg, Germany) and added 15 µL 0.01% methylamine. These extracts were analyzed by high-resolution Fourier transform mass spectrometry (FTMS) in negative ion mode using either an LTQ Orbitrap XL or an Orbitrap Fusion Tribrid (Thermo Scientific, San Jose, CA, USA), both equipped with a Triversa NanoMate robotic nanoflow ion source (Advion Biosciences, Ithaca, NY, USA). On the LTQ Orbitrap XL, each sample was analyzed for 5 min, using an FTMS scan range of *m*/*z* 150–420, a max injection time of 250 ms, automated gain control for an ion target of 1e5, two microscans, and target resolution setting of 100,000. On the Orbitrap Fusion Tribrid, each sample was analyzed for 5 min, using a FTMS scan range of *m*/*z* 150–420, a max injection time of 100 ms, automated gain control for an ion target of 1e5, three microscans, and target resolution setting of 500,000. The sample injection settings of the NanoMate for samples injected with ammonium formate were −0.96 kV and 1.25 psi. The NanoMate settings for samples injected with methylamine were −0.96 kV and 0.6 psi.

### 2.5. Lipid Identification and Quantification

NEFA analytes were identified and quantified using the ALEX^123^ software framework [18,29]. Mass spectral searches were carried out using a mass tolerance of ±0.001 amu. Reported NEFA species are denoted by sum composition and quantified by normalizing their intensities to the intensity of the internal standard and multiplication by its spike amount.

## 3. Results and Discussion

### 3.1. Specific Detection of 13C-Labeled NEFA Produced In Vitro by FASN

The aim of this study was to develop a direct FASN activity assay supporting specific, precise, and accurate monitoring of the specific activity and the product-spectrum of FASN. To this end, we made use of FASN purified from cow mammary gland, incubated this with the reactants ^13^C_3_-malonyl-CoA, acetyl-CoA, and NADPH at 37 °C for 10 min, and initially performed total FA analysis using high-resolution shotgun lipidomics in negative ion mode using the workflow described previously by Gallego et al. [26]. This analysis revealed specific detection of several ^13^C isotopologues of the canonical FASN product NEFA 16:0, having from 12 to 16 ^13^C atoms (Figure 1A,B). Of these, NEFA 16:0(+^13^C_14_) (*m*/*z* 269.2800, −0.4 ppm mass accuracy) and NEFA 16:0(+^13^C_16_) (*m*/*z* 271.2868, −0.7 ppm mass accuracy) were the most abundant isotopologues, representing 53% and 40%, respectively, of all *de novo* synthetized NEFA 16:0. The analysis also revealed detection of other ^13^C-labeled NEFAs, including NEFA 12:0, 14:0, 18:0, and 20:0. In general, all the identified ^13^C-labeled NEFAs had a characteristic isotope pattern where the most abundant isotopologue had n-2 ^13^C atoms (where n denotes the total number of C atoms in a particular NEFA analyte) followed by the fully ^13^C-labeled NEFA analyte (i.e., ^13^C_n_). At first this was surprising given that the reaction mechanism of FASN (Appendix A) predicts synthesis of primarily the ^13^C_n-2_ isotopologue since unlabeled acetyl-CoA is used as a primer. However, FTMS analysis of the commercially available ^13^C_3_-malonyl-CoA revealed that this reactant is contaminated with ^13^C_2_-acetyl-CoA (Appendix A), which probably results from decarboxylation of the labile ^13^C_3_-malonyl-CoA. As such, under the used assay conditions FASN can use both unlabeled and ^13^C-labeled acetyl-CoA as primers and thereby produce both the ^13^C_n-2_ and ^13^C_n_-labeled NEFAs, respectively. We note that omitting the ^13^C_3_-malonyl-CoA from the reaction mixture resulted in no production of ^13^C-labeled NEFAs (Figure 1A), which demonstrates that the high-resolution mass spectrometric analysis, and the assay, is essentially noise-free and specific. Finally, to estimate the amount of *de novo* synthetized NEFAs the intensities of different isotopologue belonging to different NEFA analytes were summed and normalized to the intensity of the added internal NEFA 19:0 standard and multiplied with the its spike amount. This showed that under the applied assay conditions 0.02, 0.6, 21, 4, and 0.2 pmol of ^13^C-labeled NEFA 12:0, 14:0, 16:0, 18:0, and 20:0, respectively, were produced after 10 min (Figure 1G).

### 3.2. Optimized Workflow for MS-Based Analysis of FASN Activity

Having established that our recently reported method for total FA analysis [26] also affords specific monitoring of *de novo* synthetized ^13^C-labeled NEFAs, we next evaluated whether we could simplify parts of the workflow for analysis of FASN activity. Notably, our original method was designed for total FA profiling of complex sample matrices, which requires lipid hydrolysis to liberate FA chains for subsequent mass spectrometric analysis. However, given that the end products of FASN are NEFAs (Appendix A) we deemed that the hydrolysis step could be omitted. Furthermore, we also rationalized that the highly sensitive and specific analysis of deprotonated ^13^C-labeled NEFA analytes could potentially allow us to omit the commonly used sample evaporation step where extracts are concentrated prior to being resuspended in a low volume of solvent that is compatible with the subsequent mass spectrometric analysis.

To determine whether these time-consuming, more labor-intensive, and seemingly redundant sample preparation steps could be omitted, we carried out a systematic test of four different sample preparation strategies (Figure 2): (i) the approach used for total FA profiling with acid-catalyzed hydrolysis, extraction with *n*-hexane, followed by sample evaporation and resuspension in chloroform/methanol (1:2, *v*/*v*) prior to analysis (as in Figure 1); (ii) the approach for total FA profiling but without acid-catalyzed hydrolysis; (iii) a Bligh and Dyer [28] based approach where NEFAs are extracted with chloroform/methanol (2:1, *v*/*v*), followed by sample evaporation and resuspension in chloroform/methanol (1:2, *v*/*v*) prior to analysis; and (iv) the Bligh and Dyer-based approach where the extract is not evaporated but is simply directly injected for FTMS analysis. Overall, this comparison showed that the amounts of quantifiable *de novo* synthetized ^13^C-labeled NEFA were essentially the same and independent of the applied sample preparation strategy (Figure 2). Based on this we concluded that both the acid-catalyzed hydrolysis (as expected) and the sample evaporation can be omitted for analysis of FASN activity. Furthermore, for the simplicity of the workflow we decided to make use of the Bligh and Dyer-based approach where NEFA analytes are extracted with chloroform/methanol (2:1, *v*/*v*) and directly analyzed by high-resolution FTMS analysis, given this is more time-efficient and less hands-on, as compared to the other procedures. Overall, this consolidated a relatively simple sample preparation routine that only requires spiking the reaction mixture with an internal standard, adding chloroform/methanol, vigorously mixing the sample, collecting an aliquot of extract, and loading it in a 96-well plate for subsequent automated sample injection and FTMS analysis.

### 3.3. Determination of Specific FASN Activity

Determining the specific activity of FASN requires monitoring the initial rate of the FASN-catalyzed reaction and therefore knowledge about how this enzyme reaction evolves over time. Specifically, the initial rate is calculated using data where the rate of product formation is linear with time and all substrates are at saturating levels [30]. In order to determine the time interval, for which the formation of ^13^C-labeled NEFAs are linear with time, we first carried out a time series analysis of the FASN-catalyzed reaction over a prolonged time interval of 90 min (Figure 3A). This analysis showed that the production of ^13^C-labeled NEFAs is only linear within the first 5 min and afterwards gradually reaches a plateau. Based on this, we decided to monitor the initial rates of the FASN-catalyzed reaction by time series analysis across the first 2 min with sampling every 20 s (Figure 3B–D). Analyzing such data by linear regression and determining the slope value and normalizing to the amount of FASN we estimated the specific activity for production of NEFA 14:0, 16:0, and 18:0 to be 0.11, 3.3, and 0.52 pmol NEFA/min/µg FASN, respectively (Figure 3E). As expected, this showed that the specific activity for production of the canonical NEFA 16:0 was at least 6.4-fold higher than that of the other FASN products. Collectively, the total specific activity of the FASN, based on the sum of individual activities, was 3.95 pmol NEFA/min/µg FASN (Figure 3E). In terms of NADPH equivalents, as determined by the conventional spectrophotometry-based assay, this value corresponds to a specific activity of 56.1 pmol NADPH oxidized/min/µg FASN (Figure 3E).

## 4. Conclusions

Here, we described a novel assay that supports swift, specific, precise, and accurate determination of FASN activity in vitro. The method builds on the classical activity assay where purified FASN is incubated with acetyl-CoA, NADPH and, in our setup, ^13^C-labeled instead of unlabeled malonyl-CoA. This is followed by spiking the reaction mixture with an internal NEFA standard, performing lipid extraction with without the conventional sample evaporation step and instead simply analyzing the crude extract by automated high-resolution FTMS analysis for quantitative monitoring of *de novo* synthetized ^13^C-labeled NEFAs. We demonstrate that our method is specific and sensitive, with detection of labeled NEFAs at the femtomolar level and the ability to provide in-depth characterization of the FASN product-specificity. As such, we show that purified cow FASN not only produces the canonical NEFA 16:0 (i.e., palmitic acid) but also other, less abundant NEFAs having chain-lengths between 12 and 20 carbon atoms. Finally, we demonstrate, using time series analysis, that our methodology allows determining the specific activity of FASN for producing NEFAs with different chain-lengths. Overall, we carried out the analysis in roughly 70 min per sample, with approximately 60 min for sample extraction, 5 min for FTMS analysis, and the remainder for sample handling and pipetting. We note that more optimization, and the use of robotics, could potentially reduce the overall time to about 5 min per sample, by shortening the extraction step to about 3 min, the FTMS to 1 min (as we have previously reported [26]), and the remaining time for sample handling and pipetting. Finally, we deem that our assay could be highly useful for laboratories equipped with commonly available high-resolution mass spectrometers that have a resolving power higher than ≈95,000 (full width at half maximum at *m*/*z* 300) [26]. Moreover, the assay can also be carried out using time-of-flight-based instrumentation, but would potentially require use of deisotoping algorithmics to reduce bias from peak coalescence attributed chemical background noise.

In comparison to other approaches our novel assay strikes a unique balance between the ease of operation and the analytical performance. On one hand our assay rivals the analytical simplicity of the classical and indirect spectrophotometry-based approach that monitors NADPH consumption [19,31,32] by enabling direct and specific detection of NEFA analytes in crude extracts, eliminating the need for sample evaporation/concentration and dissolving extracts in an MS-compatible solvent. On the other hand, our assay rivals the analytical performance of previously reported GC-MS-based approaches [23,24] by not requiring an additional step to derivatize NEFA analytes and using high-resolution FTMS analysis with essentially noise-free detection and simultaneous quantification of individual ^13^C-labeled NEFAs. We argue that these analytical hallmarks, together with additional method optimization, could transform our assay into a high-throughput screening technology for developing potent FASN inhibitors for combating several human diseases [33,34] and for enzyme engineering aimed at modulating the activity and the product-species of fatty acid syntheses in biotechnological settings. Finally, we note that our assay can also be carried out using other CoA-activated metabolites, including the branched chain amino acid-derived metabolites propionyl-CoA, isobutyryl-CoA, isovaleryl-CoA, and 2-methylbutyryl-CoA, to help gain insights into the molecular underpinning of how FASN enzyme promiscuity drives branched-chain FA synthesis in adipose tissues and contributes to type-2-diabetes and other metabolic diseases [35,36].

## Figures and Tables

**Figure 1 biomolecules-10-00118-f001:**
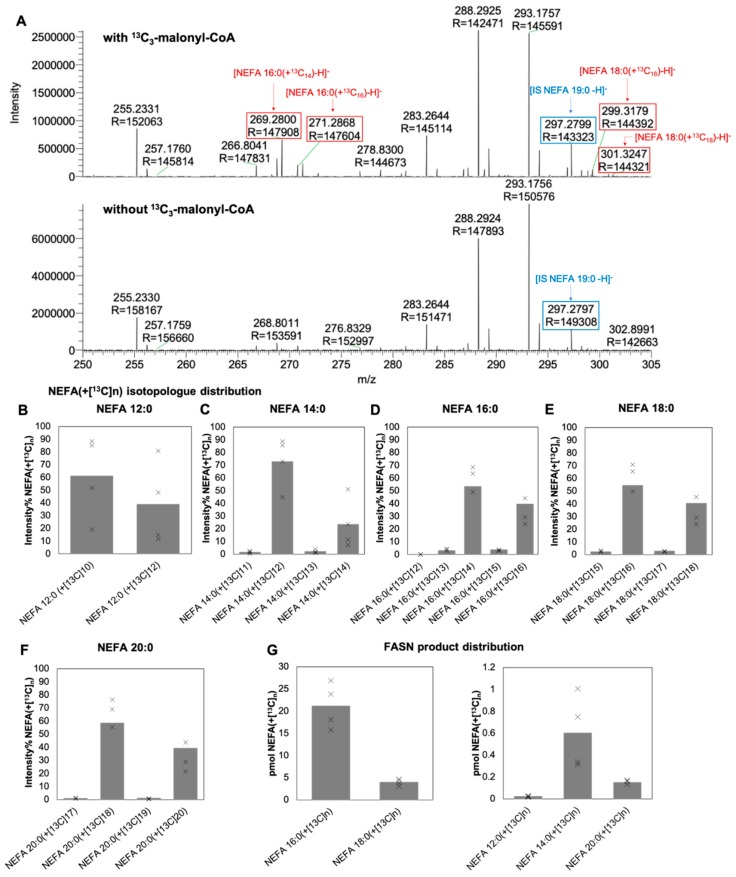
Specific monitoring of *de novo* synthetized ^13^C-labeled NEFAs. (**A**) Negative ion mode Fourier transform mass spectrometry (FTMS) analysis of deprotonated ^13^C-labeled NEFA analytes. Purified fatty acid synthase complex (FASN) was incubated with ^13^C_3_-malonyl-CoA, acetyl-CoA and NADPH (top panel) or acetyl-CoA and NADPH (lower panel) at 37 °C for 10 min. Subsequently the reaction mixtures were subjected to total fatty acyl (FA) analysis [26] using acid-catalyzed hydrolysis in the presence of internal 19:0-CoA standard, followed by extraction with hexane and FTMS analysis using an LTQ Orbitrap XL mass spectrometer. Ions representing the internal standard are indicated in blue, while detected ^13^C-labeled NEFAs are indicated in red. The abundant ions at *m*/*z* 288.2925 and 293.1757 are due to non-interfering chemical background noise. (**B**–**F**) ^13^C isotopologue distribution of indicated NEFA species. Data represent average values and dots individual measurements (*n* = 4 independent reactions extracted and analyzed separately). (**G**) Average pmol of *de novo* synthetized ^13^C-labeled NEFA per µg FASN per 10 min. Data represent average values and dots individual measurements (*n* = 4 independent reactions extracted and analyzed separately).

**Figure 2 biomolecules-10-00118-f002:**
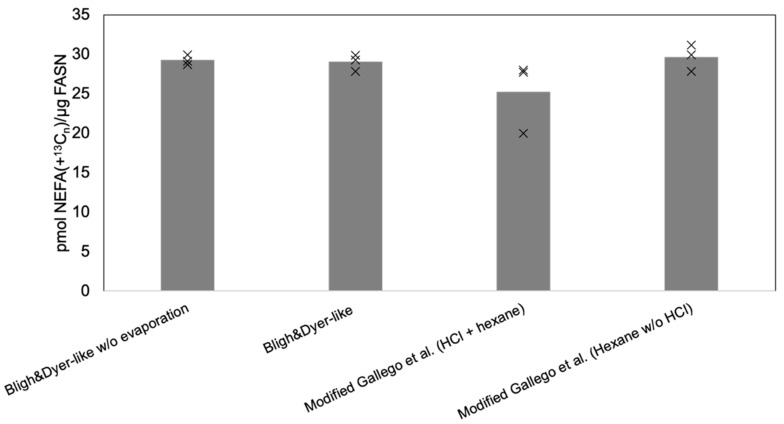
Optimizing the workflow for monitoring FASN activity in vitro. Purified FASN was incubated with ^13^C_3_-malonyl-CoA, acetyl-CoA and NADPH at 37 °C for 10 min. The reaction mixtures (200 µL), with *de novo* synthetized ^13^C-labeled NEFAs, were processed using one of the four indicated strategies (see the Results and Material and Methods sections for details). Deprotonated NEFAs were detected by negative ion mode FTMS using an LTQ Orbitrap XL mass spectrometer and quantified using the internal NEFA 16:0(+^2^H_4_) standard and normalized to the amount of FASN. Data represent average values and dots individual measurements (*n* = 3 independent reactions extracted and analyzed separately).

**Figure 3 biomolecules-10-00118-f003:**
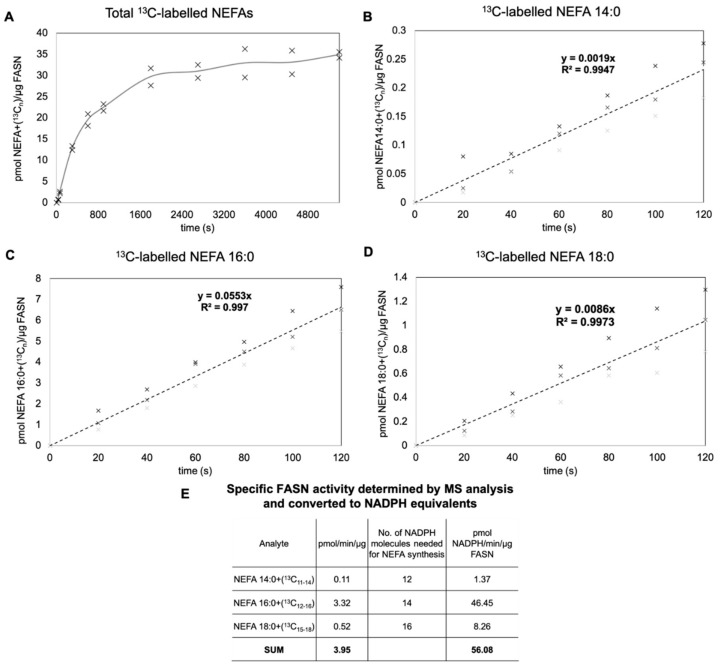
Determination of specific FASN activity. (**A**) Time series analysis of total *de novo* synthetized ^13^C-labeled NEFAs across a 90 min period. Purified FASN was incubated with ^13^C_3_-malonyl-CoA, acetyl-CoA, and NADPH at 37 °C for the indicated times. The reaction mixtures were extracted using a Bligh and Dyer-based approach without sample evaporation. Extracts were analyzed directly by FTMS on an Orbitrap Fusion Tribrid mass spectrometer and ^13^C-labeled NEFAs were quantified using the internal standard NEFA 16:0(+^2^H_4_). Data represent average values and dots individual measurements (*n* = 2 independent reactions extracted and analyzed separately). (**B**–**D**) Time series analysis of *de novo* synthetized ^13^C-labeled NEFA 14:0, 16:0, and 18:0 across a 2 min period, representing the initial rate of the FASN-catalyzed reaction. The assay was carried out as outlined in (A). Data represent average values and dots individual measurements (*n* = 3 independent reactions extracted and analyzed separately). Lines represent linear regressions having the indicated slope values and regression coefficients. (**E**) Summary of estimated specific FASN activity for different NEFA species and calculation of total specific activity in terms of NEFAs and NADPH equivalents.

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
