# Peer review of "A Simple and Direct Assay for Monitoring Fatty Acid Synthase Activity and Product-Specificity by High-Resolution Mass Spectrometry"

_biomolecules, 2020, doi:10.3390/biom10010118_

Round 1

Reviewer 1 Report

In my opinion the experiment was well designed, performeed and described. The manuscript provides valuable and interesting information.

I have some mino remarks presented below.

Subsection 2.1 Please state what was quality of reagents. I suppose that MS grade, but it should be stated.

Subsection 2.2. Is protocol of FASN purification published? If not it should be cited as "unpublished".

Discussion. Authors used Orbitrap mass spectrometer. Is there possible to dicuss briefly perspective for transfer method into another kind of mass spectrometer (such as Q-ToF or tripe quadrupol)? what are possible problems?

References
Please check reference list and correct if necessary. For instance ref. No 25 (Gallego et al. 2019) should be cited as Biomolecules 2019, 9, Article no 7

Author Response

Please see the enclosed file: "200105_Response_to_reviewers_comments.pdf"

Reviewer 2 Report

Topolska and coworkers reported an assay to determine fatty acid synthase activity. The assay used high resolution mass spectrometry to qualitatively and quantitively measure the activity of type I fatty acid synthase complex (FASN) by monitoring its analytical product, non-esterified palmitic acid (NEFA 16:0). The authors compared four different sample preparation methods and concluded that while no significant difference was observed, the Bligh & Dyer-based approach where NEFA analytes were extracted with chloroform/methanol (2:1, vol/vol) and directly analyzed by MS was the most convenient method. Using purified FASN with 13C3-malonyl-CoA, acetyl-CoA and NADPH, the authors showed that in addition to the canonical NEFA 16:0, NEFA 12:0, 14:0, 18:0 and 20:0 were also produced. Overall, the manuscript was well written and straightforward. However, I have some questions that should be addressed before the manuscript is accepted for publication.

The first question is the identity of the most abundant two ions, ie, m/z 288 and 293. What are these two ions? Since the sample preparation was done using hexane or chloroform, it is reasonable to speculate that they are fatty acids or hydrophobic compounds.

The second question is why their intensities are different in sample without the labelled malonyl CoA (lower panel) but the same in sample with the labelled malonyl CoA (upper panel)? This discrepancy makes one doubt the quantification results.

The third question is about the isotopologue distribution as shown in Figure 1 B - F. The authors should briefly explain why and how the isotopes were distributed differently in the same NEFA.

Author Response

(The authors gave the same response as above.)
